# Unravelling the room-temperature atomic structure and growth kinetics of lithium metal

Chao Liang 1, Xun Zhang[1], Shuixin Xia[1], Zeyu Wang[1], Jiayi Wu[1], Biao Yuan 1, Xin Luo[1], Weiyan Liu[1], Wei Liu 1 & Yi Yu 1✉

Alkali metals are widely studied in various fields such as medicine and battery. However, limited by the chemical reactivity and electron/ion beam sensitivity, the intrinsic atomic structure of alkali metals and its fundamental properties are difficult to be revealed. Here, a simple and versatile method is proposed to form the alkali metals in situ inside the transmission electron microscope. Taking alkali salts as the starting materials and electron beam as the trigger, alkali metals can be obtained directly. With this method, atomic resolution imaging of lithium and sodium metal is achieved at room temperature, and the growth of alkali metals is visualized at atomic-scale with millisecond temporal resolution. Furthermore, our observations unravel the ambiguities in lithium metal growth on garnet-type solid electrolytes for lithium-metal batteries. Finally, our method enables a direct study of physical contact property of lithium metal as well as its surface passivation oxide layer, which may contribute to better understanding of lithium dendrite and solid electrolyte interphase issues in lithium ion batteries.

[1] School of Physical Science and Technology, ShanghaiTech University, Shanghai 201210, China. ✉email: yuyi1@shanghaitech.edu.cn

It has been a long history for the research on alkali metals[1], the most excitement it has generated in recent years is for energy conversion and storage[2-6]. As the "holy grail", lithium metal is considered to be the ultimate choice of anode in battery as it has the highest theoretical capacity and lowest electrochemical potential[2,3]. However, for alkali metals, less is known about their structural information on the atomic-level. An important reason is that they are so reactive and can never exist in air in elemental form. So far, vacuum or cryogenic transfer seems to be the only reliable way to take alkali metal samples into the microscopes for observation. However, contaminations and degradations cannot be entirely ruled out during the transfer process. Moreover, the electron beam sensitive nature of alkali metals disables high-resolution transmission electron microscopy (HRTEM) imaging. Until recently, atomic-scale imaging of cryo-transferred lithium has been reported at low temperature[7,8], and room temperature imaging is achieved by intercalating lithium into graphene sheets[9], whereas a direct room temperature atomic resolution imaging of bare alkali metals has never been demonstrated. More importantly, direct in situ observation of the growth of alkali metals at high spatiotemporal resolution has not been achieved yet. The missing link in the microstructure of lithium metals hinders further understanding and development of lithium-ion batteries.

In the present work, we propose a simple and versatile method to form alkali metals inside the transmission electron microscope (TEM) directly. Taking the observing electron beam as the trigger, alkali metals can be produced from their corresponding alkali salts. Here, two types of alkali salts, i.e., alkali carbonates and alkali fluorides, are chosen as demonstrations. Both alkali carbonates and fluorides are alkali compounds used in a wide field of industrial, technical and medical applications[1,10-14]. Taking $Li_2CO_3$ as an example, it is the first chemical in the lithium production chain and also the key component in lithium-air batteries[11,12]. In addition, it is an essential medicine for the treatment of manic depression and bipolar disorder[13]. Moreover, both $Li_2CO_3$ and LiF are the starting materials for preparing of various ceramics, glass and cements[1] in broad applications. Therefore, fundamental research on the stability and decomposition of lithium carbonates and fluorides can also be interesting in general.

For the electron beam-induced decomposition reaction observed here, we found pure alkali metals as the products. Compared to the conventional multistep reactions to produce solid alkali metals[15,16], the observed one-step reaction here, on the other hand, provides an elegant way to prepare and study the intrinsic structure and structural kinetics of alkali metals. During observation, the electron beam damage is minimized and atomic resolution is obtained by using low dose aberration-corrected HRTEM (AC-HRTEM)[17]. Atomic resolution imaging of lithium and sodium is achieved at room temperature. In situ fast camera enables millisecond temporal resolution so that the process of the alkali metal growth can be tracked. This work provides a paradigm to investigate such chemically reactive and electron beam sensitive materials. The immediate advantage is that an ambiguity about lithium metal growth on garnet-type solid electrolytes for lithium-metal batteries can be clarified, and physical contact property of lithium metals and surface passivation oxides can be revealed.

## Results

**In situ formation of alkali metals from alkali salts**. Figure 1a shows the schematic of our configuration. The reaction can be triggered by the observing electron beam while the reaction rate is controlled by the intensity of the irradiation. The key point is

to control the dose-rate of the electron beam. Typically, $10{\sim}1000\,e\,Å^{-2}\,s^{-1}$ is a good choice. By focusing the high energy electron beam onto the edge of alkali carbonate/fluoride particles, alkali metal particles grow out from the illuminated spot and scale to hundreds of nanometers. The reaction rate is carefully controlled by tuning the beam intensity so that the reaction process is slow enough to be captured by the detector, and atomic resolution imaging of both lithium and sodium is demonstrated at room temperature. Details are provided in the following. To begin, pure alkali carbonates/fluorides were drop-casted onto the TEM grids. The basic structural and compositional characterizations of the as-drop-casted alkali salts are shown in Supplementary Figs. 1–3, confirming pure-phase alkali salts acting as the starting materials. Next, three examples of the alkali metal formation and growth process are demonstrated. Figure 1b–d show the selected sequential snapshots of in situ growth process of lithium from lithium carbonate (Supplementary Movie 1, ${\sim}100\,e\,Å^{-2}\,s^{-1}$), sodium from sodium carbonate (Supplementary Movie 2, ${\sim}10\,e\,Å^{-2}\,s^{-1}$) and lithium from lithium fluoride (Supplementary Movie 3, ${\sim}20\,e\,Å^{-2}\,s^{-1}$), respectively. During the growth process, sharp edges and corners can be observed which indicate the crystalline feature of these particles. In general, the particles would expand along several growth directions (denoted by white arrows). It is concluded that the electron beam-induced formation of alkali metals from alkali salts could be a common phenomenon.

In order to confirm the composition and structure of the particles, selected area electron diffraction (SAED) was performed on lithium and sodium particles, respectively. Figure 2a shows two neighbor particles grown from lithium carbonates and the black circle represents the selected area for diffraction. Figure 2b, c shows the SAED pattern and its rotational average spectrum, respectively. Figure 2d, g shows the crystal structures of body-center cubic (b.c.c.) lithium and face-center cubic (f.c.c.) lithium oxide, respectively. Their corresponding simulated polycrystalline diffraction patterns and rotational average spectra are shown in Fig. 2e, f, h, i, respectively. By comparison, it can be concluded that the particles are comprised of both lithium and lithium oxide. To be noted that the polycrystalline-featured rings are indexed to be lithium oxide while the isolated sharp single-crystalline diffraction spots are indexed to be lithium. Similar analysis for the sodium case is shown in Fig. 2j–r. Hence, these particles are composed of single-crystalline alkali metals and their polycrystalline oxide compounds.

**Growth kinetics of lithium metal**. To understand the formation sequence of alkali metals and their oxides, the growth of particles was traced at high spatiotemporal resolution[18]. Here, combining low dose AC-HRTEM imaging with in situ fast camera detection, an example of lithium-particle growth is demonstrated at atomic resolution with fast frame rate (40 ms per frame). It is confirmed that pure-phase lithium metal was formed and grew up at the beginning. Figure 3a shows an AC-HRTEM image of a lithium particle along [111] zone axis. The b.c.c. structure can be verified from the Fourier transformation pattern. The surface of two exposed <110> facets can be seen clearly. An in situ movie (Supplementary Movie 4 and snapshots shown in Fig. 3) indicates that the lithium particle grew along these two directions one after another. Figure 3a–c depicts the growth stage along [10$\bar{1}$] direction, as indicated by red arrows. Along this direction, (10$\bar{1}$) facet moved forward while (1$\bar{1}$0) facet remained unchanged. Evolution of the (10$\bar{1}$) edge is highlighted (red single solid lines to dotted lines, and then to double solid lines), the layer-by-layer growth of lithium atoms can be observed. This stage lasted for about 600 ms (15 frames) and later on grew slowly for three seconds, then the

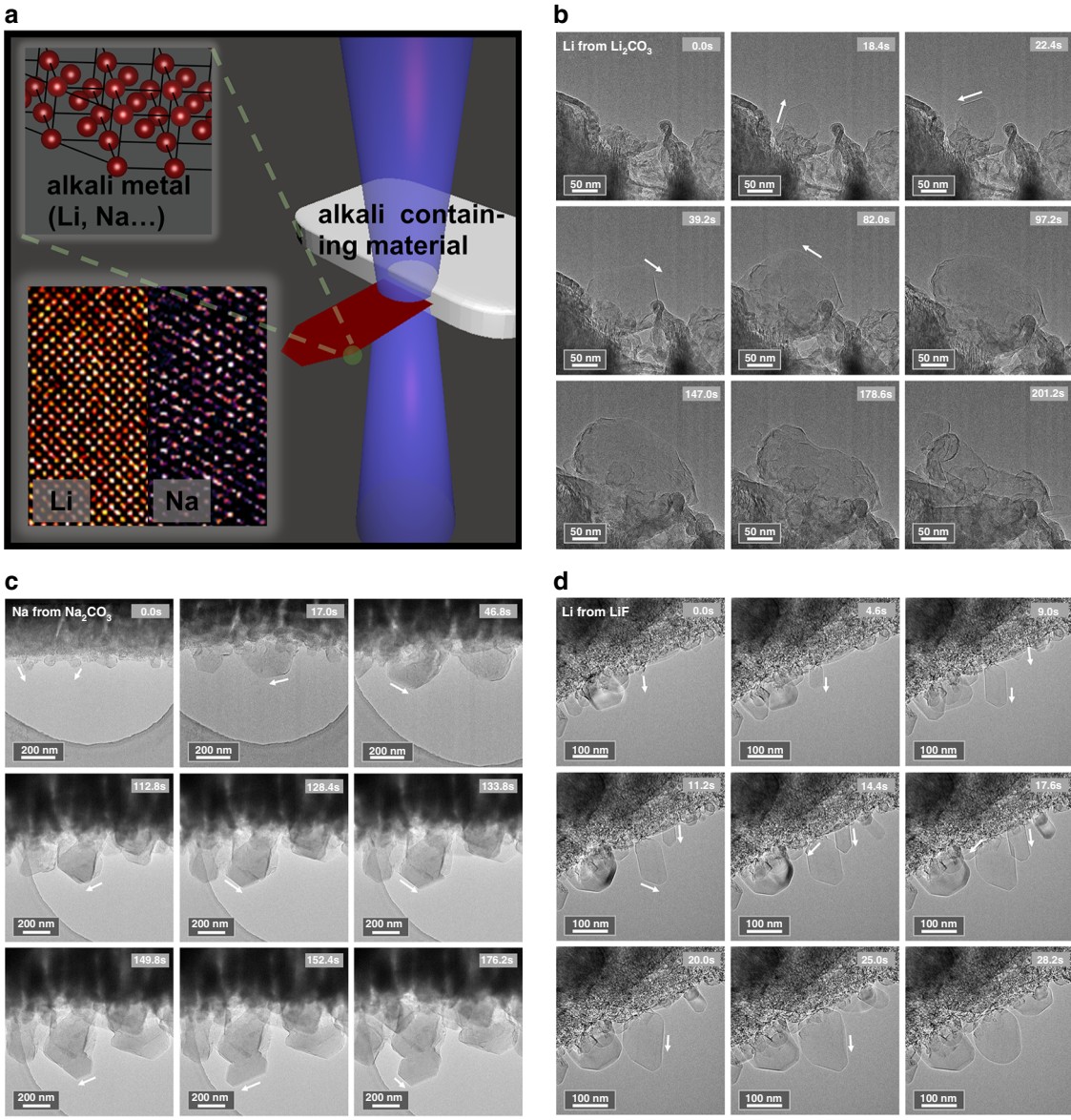

**Fig. 1 In situ formation and growth of alkali metal particles. a** Schematic of the experimental process. An alkali (Li, Na) metal particle formed from their corresponding alkali containing materials under beam irradiation. The inset shows the b.c.c. structure of alkali metals. Atomic resolution images of both lithium and sodium metals are shown at room temperature condition. **b** In situ growth of Li particles from lithium carbonate. **c** In situ growth of Na particles from sodium carbonate. **d** In situ growth of Li particles from lithium fluoride.

particle changed growth direction to [1$\bar{1}$0], as shown in Fig. 3d–f. Along this direction, (1$\bar{1}$0) facet moved forward as indicated by red arrows. Compared with the (10$\bar{1}$) edge, growth along the [1$\bar{1}$0] direction is rather quick. Hence, atoms at the corner could not be replenished in a short time, and an incomplete (2$\bar{1}\bar{1}$) facet was exposed as a transient state. As time went on, lithium atoms were replenished gradually, and the corner surface changed to even high index facets and the length of this corner shorten from 14 nm to 6 nm, resulting in a sharp corner eventually (also see Supplementary Movie 4). This stage lasted for about 400 ms. For alkali metals with b.c.c. structure, <110> face is the most densely packed face and has lowest surface energy, therefore the surface ended up with low index facets again. Millisecond temporal resolution enables the observation of fast structural evolution, confirming that the surface energy is one of the driving forces for the growth of alkali metals.

Figure 3g, h shows the growth length and growth rate (Δlength/Δt) of the particle versus time, respectively. Particle

length at $t = 0$ is regarded as the zero point. Arrows mark each start of a new growth period. As can be seen in Fig. 3g, in each growth period, the particle grows quickly at first and this rate slows down gradually, and therefore growth steps form. It is noted that the growing process is not continuous, and it may be possible that the delivery of new lithium source from alkali salts takes time for the start of the next growing process. After the change of growth direction, the growth along new direction is rather fast. As shown in Fig. 3h, growth rate at this moment (the blue arrow) is almost one order of magnitude quicker than the previous fastest growth rate (the leftmost red arrow). This indicates a sudden influx of a large number of lithium atoms or possible accumulation of a large number of diffused lithium atoms before the change of the growth direction. In terms of the different growth rate between [10$\bar{1}$] and [1$\bar{1}$0] direction, it is related with the anisotropic mass transfer along these two directions[19]. The rate of mass transfer could be determined by the decomposition of the matrix of the lithium salts. Anisotropic

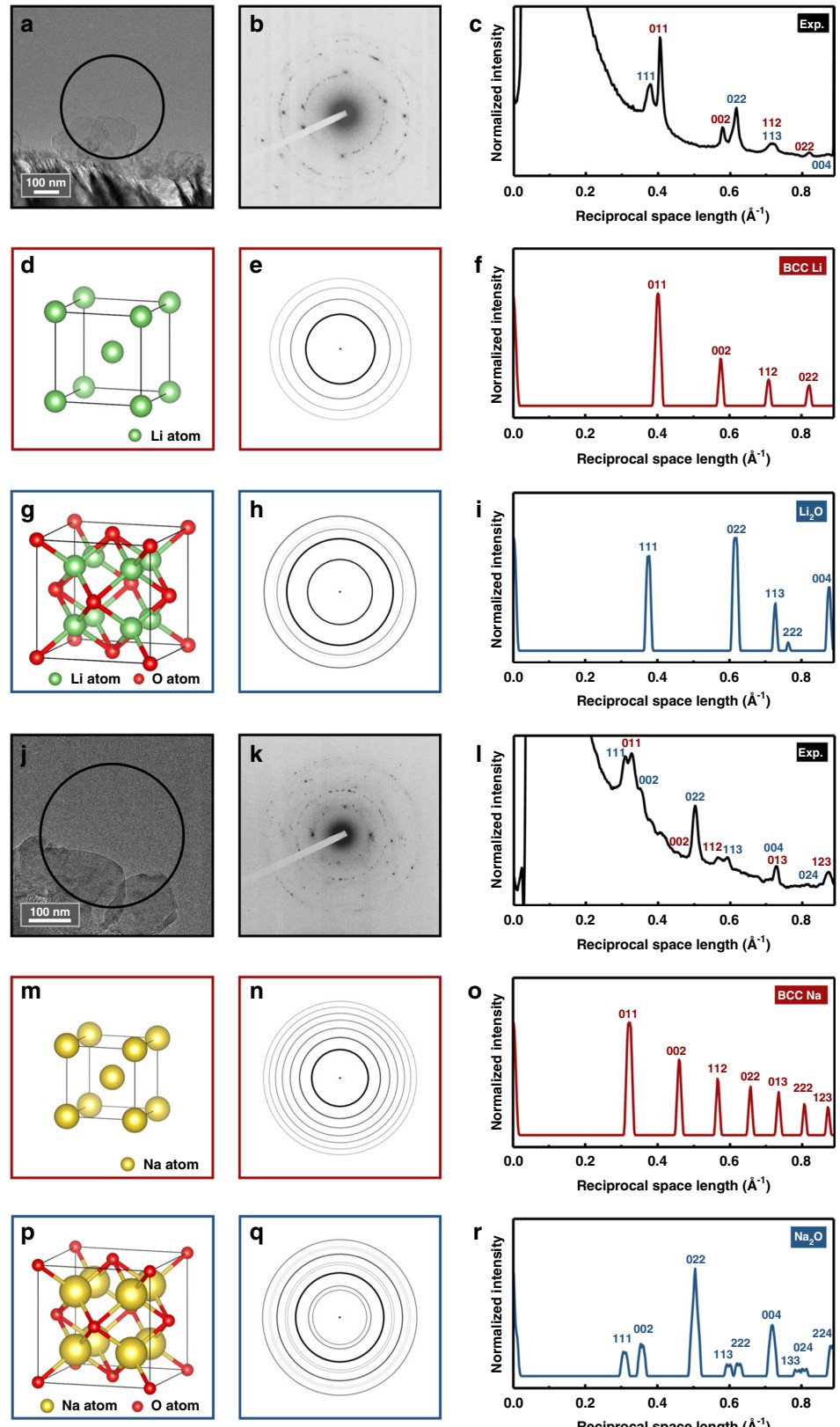

**Fig. 2 SAED analysis of alkali metal particles. a** The selected lithium particles (the black circle represents the aperture area), **b** its SAED pattern and **c** rotational average spectrum. **d** Crystal structure of b.c.c. lithium, **e** its simulated polycrystalline diffraction rings and **f** rotational average spectrum. **g** Crystal structure of f.c.c. lithium oxide, **h** its simulated polycrystalline diffraction rings and **i** rotational average spectrum. **j** The selected sodium particles, **k** its SAED pattern and **l** rotational average spectrum. **m** Crystal structure of b.c.c. sodium, **n** its simulated polycrystalline diffraction rings and **o** rotational average spectrum. **p** Crystal structure of f.c.c. sodium oxide, **q** its simulated polycrystalline diffraction rings and **r** rotational average spectrum.

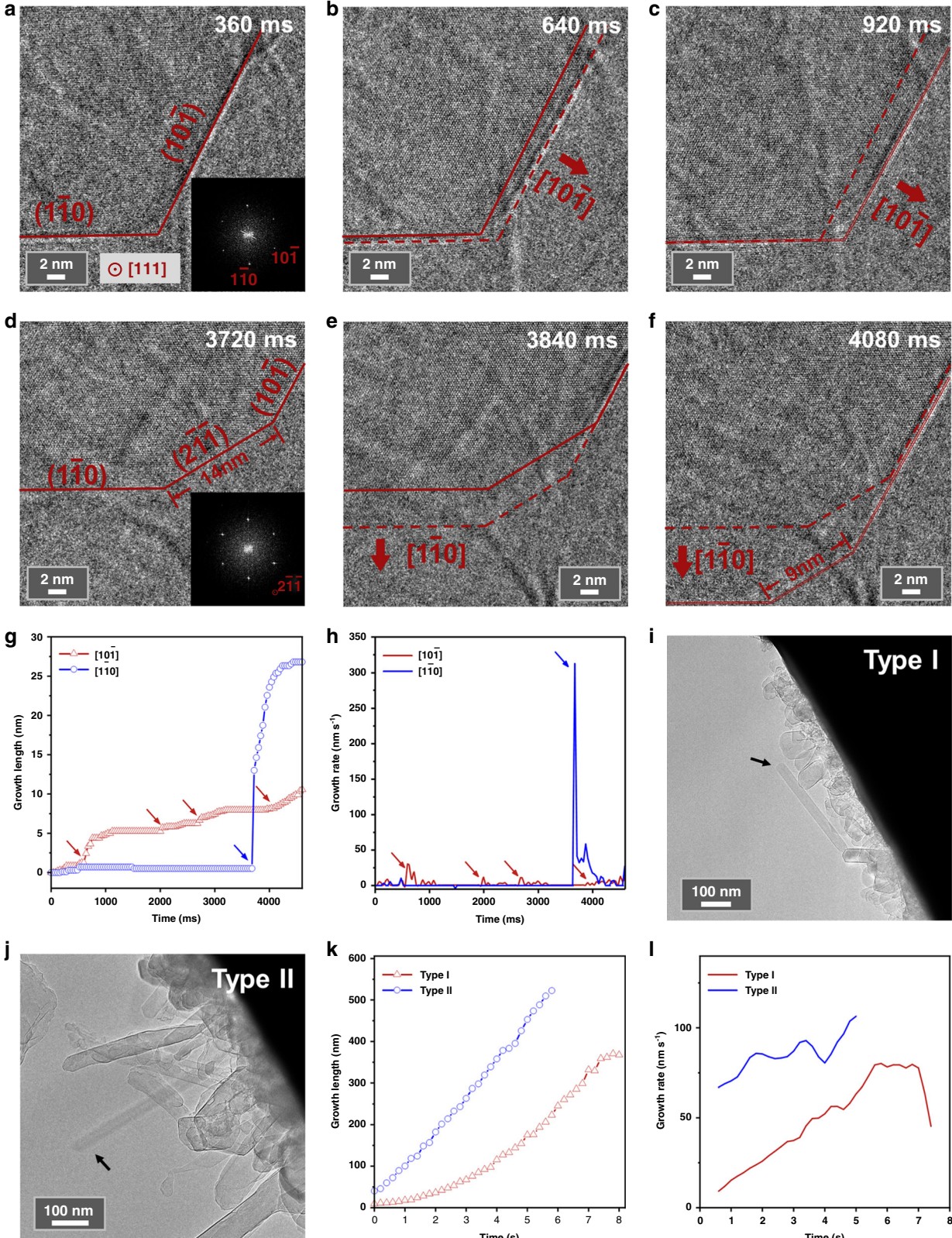

decomposition of the lithium salts may provide anisotropic diffusion flux of lithium source along different directions.

Apart from the growth of lithium particles, we have also observed the growth of lithium dendritic whiskers. Investigation of the whisker growth here can be compared with the electrical current-driven growth of lithium dendrites in lithium-ion batteries[20], which may provide better understanding of the dendrite growth. Figure 3i, j shows the growth of lithium whiskers from an as-synthesized lithium particle (defined as Type I, Supplementary Movie 5) and from the root lithium fluoride materials (defined as Type II, Supplementary Movie 6), respectively. Their corresponding growth length and smoothed growth

**Fig. 3 Growth kinetics of lithium metal. a–c** Growth along [10$\bar{1}$] direction. **a** A [111]-oriented lithium particle grown from lithium fluoride. Inset is the corresponding Fourier transformation pattern. Red single solid lines represent the edge of this particle. Image filtering was applied to enhance the signal to noise ratio. Dose-rate is ~1000 e Å$^{-2}$ s$^{-1}$. **b** Edge of the particle at this stage are tracked by red dotted lines. The middle area between solid lines and dotted lines represents the additive grown portion. **c** Edge of the particle at this stage are tracked by red double solid lines. The middle area between solid lines and dotted lines represents the additive grown portion. **d–f** Growth along [1$\bar{1}$0] direction. **d** Red single solid lines represent the edge of this particle. (2$\bar{1}\bar{1}$) surface is exposed and the corresponding Fourier transformation spot is marked by red circle. **e** Edge of the particle at this stage are tracked by red dotted lines. **f** Edge of the particle at this stage are tracked by red double solid lines. **g** Growth length versus time. Arrows represent starts of new growing processes. **h** Growth rate versus time. Arrows mark the growing steps as in (**g**), respectively. **i, j** Images of two types of lithium whisker growth, Type I in (**i**) and Type II in (**j**), respectively. **k, l** Growth length (**k**) and growth rate (**l**) versus time.

rate versus time are shown in Fig. 3k, l, respectively. In these two examples, the whiskers grew straightly without deflection. It is found that the growth rate of Type I positively correlated with time (the correlation coefficient is 0.769 for the raw data and 0.993 for smoothed data), and the growth kinetics can be described as:

$$L = \frac{1}{2}at^2 \qquad (1)$$

Here $L$ represents the length of the whisker, $t$ represents the time, $a$ represents the acceleration. Hence, this is an accelerated growth mode and $a = 2.56$ nm s$^{-2}$ (0.6–5.6 s) for the case in Fig.3i. In contrast to Type I, the growth rate of Type II falls in the range of 60~100 nm s$^{-1}$ and the length increases linearly with time (the correlation coefficient is 0.998). Therefore, it is a uniform growth mode and the kinetics follows:

$$L = vt \qquad (2)$$

where $v$ represents the speed. For the case shown in Fig. 3j, $v = 85$ nm s$^{-1}$ (0–5.8 s).

Uniform growth of Type II lithium whiskers suggests the decomposition reaction of lithium salts occurs uniformly, and the formation of lithium metal may be reaction-rate limited[21]. For the case of Type I, lithium whisker started to grow from an initially formed lithium particle, and the accelerated growth indicating a different mechanism with additional driving force. Interestingly, comparing with the situation of lithium whisker growth in the liquid electrolyte of lithium-ion batteries[22], we found similar growth kinetics for whiskers grown from lithium particles, although the driving force (electron beam irradiation versus electrical current) and growth environment (vacuum versus liquid) are different. In addition, for the Type I growth demonstrated here, lithium source could not be replenished from surrounding environment as the case in liquid electrolyte, and the lithium particles should be the only lithium supplier. However, no obvious collapse or contrast change of the lithium particle can be observed as shown in Fig. 3i, indicating a rapid replenishment of lithium from other surrounding particles[20]. To summarize, 1D-growth could occur at the pinhole of the surface passivation layers of as-formed lithium particles (Type I whisker) or at some unique nucleation sites of the lithium salts where the diffusion is confined (Type II whisker). The different growth kinetics of two types of whisker could be related with different diffusion barriers in lithium metal and lithium salts[23].

In general, we propose that the growth process of both particles and whiskers could be a competition between radiation damage and diffusion flux. As the particle/whisker grows, radiation damage becomes more severe and the diffusion flux is depleted so that the growth rate slows down gradually. It seems that different environments ($CO_2$[20], $N_2$[20], vacuum, and liquid[22]) have little effect on the growth kinetics of lithium metal, whereas the morphology is related to the surface passivation layers, which is sensitive to the surrounding environment. The 1D- or 3D-growth mode is on one hand related to the confinement from surface passivation layers, and on the other hand related to the diffusion flux which is dependent of the strength of the driving force (electron beam irradiation or electrical current).

Besides, the cross sections of these whiskers could be observed as deflection occurred occasionally as the whiskers became longer. One of the cases is shown in Supplementary Fig. 4 for a Type I whisker. The hexagonal cross section is again in consistent with the electrochemically deposited lithium whiskers[7]. Consequently, intrinsic features of lithium whiskers grown in vacuum shares many similarities with the situation in batteries, and therefore our method may provide an alternative way to study lithium metal-related structure problems as high spatiotemporal resolution is a major advantage over the other methods.

During the whole in situ growing process, sharp crystalline surface facets, as well as single-crystalline feature of the particle, maintained throughout, and alkali metal oxides did not show up at an early stage. However, no matter for the decomposition of carbonates or fluorides, alkali metal oxides were found in all the final SAED analysis (Fig. 2 and Supplementary Fig. 5). This indicates that for the electron beam-induced decomposition of alkali carbonates/fluorides, pure alkali metals were formed originally. Oxidation would occur in a subsequent step, which will be detailed in the following.

**Oxidation of alkali metals.** After the nucleation and fast growth process, alkali metal particles reached certain area/volume (Fig. 1) and the growth slowed down. Static imaging and spectroscopy were applied at this stage to verify the spatial distribution of the alkali metals and oxide compounds. Figure 4a shows an AC-HRTEM image (dose-rate ~1050 e Å$^{-2}$ s$^{-1}$) of a lithium particle along [110] direction. Single-crystalline feature of the particle can be seen at a glance. Figure 4b shows a close-up where the atom columns can be observed clearly. Its further magnified image in Fig. 4c agrees well with the simulated AC-HRTEM image (inset and also Supplementary Fig. 6) of lithium. Furthermore, Fig. 4d shows the elemental lithium mapping of a typical particle using energy-filtered TEM (EFTEM). Atomic resolution imaging together with compositional mapping shows that the fresh particle grows to be a single-crystalline lithium initially. Within the scope of Fig. 4a, only a little portion of the lattice fringes is presented in the form of Moiré fringes, as indicated by the blue square. The Moiré fringes arise from the overlap of $Li_2O$ with lithium (detailed analysis shown in Supplementary Fig. 7). We found that in most cases, $Li_2O$ distributed on the outer surface of the particles. Figure 4e shows several particles which have been placed in TEM vacuum chamber for tens of minutes. Its inset shows a partial close-up of the surface and the lattice of $Li_2O$ is observed. Fourier component analysis is utilized to isolate the different orientation of $Li_2O$ crystal grains, and the result is shown in Fig. 4f–h. $Li_2O$ with three different orientations were found and indicated by different colors, with their corresponding selected Fourier transformation spots shown inset. Figure 4h is the combination map with different orientations. The above AC-HRTEM

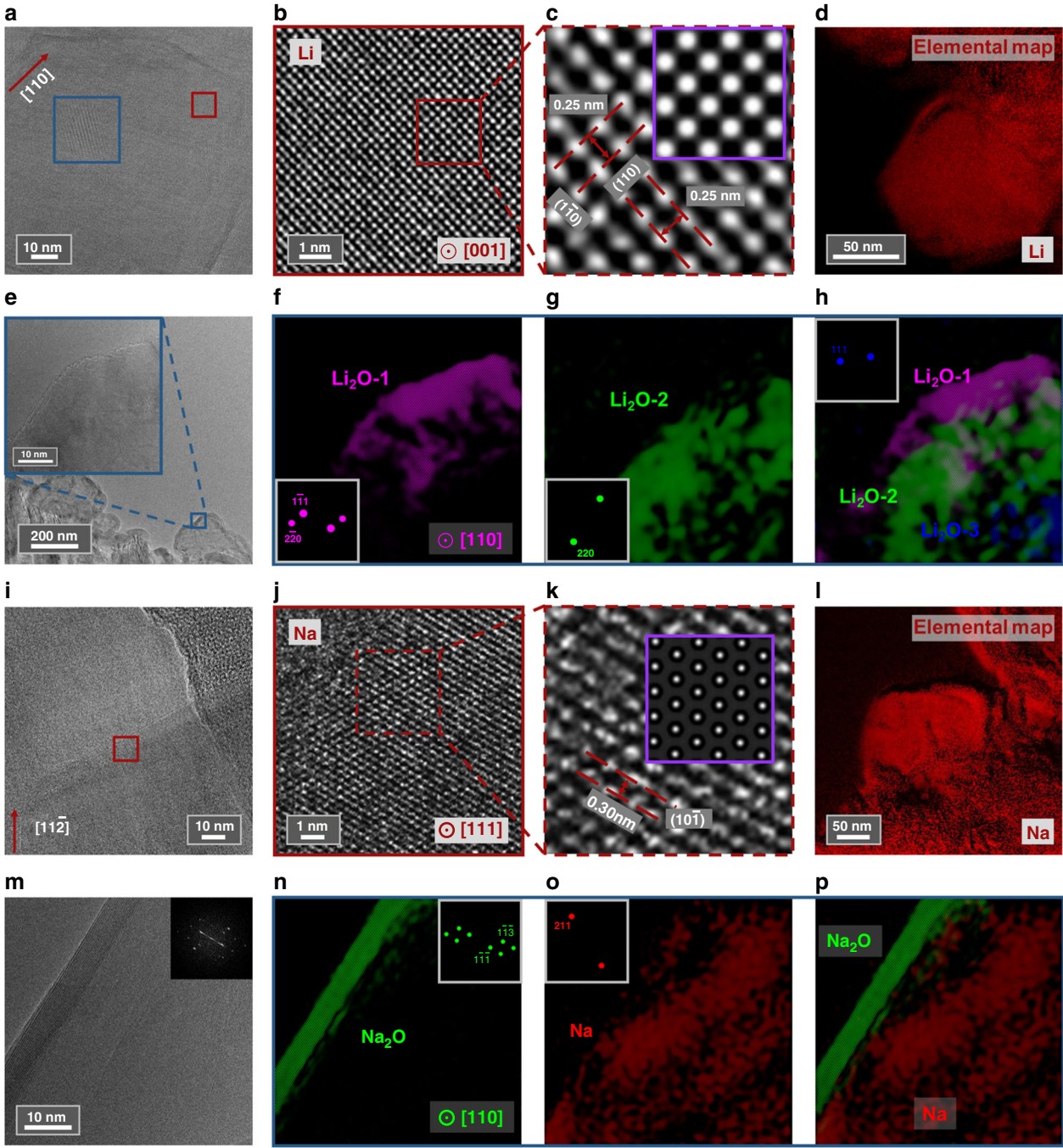

**Fig. 4 AC-HRTEM of alkali metal particles. a** A lithium particle grown along [110] direction. **b** AC-HRTEM of the particle, (**c**) partial enlarged view and simulated AC-HRTEM of lithium as inset. **d** EFTEM of a lithium particle. **e** Oxidized lithium particles and partial enlarged view as inset. **f, g** Lithium oxide with different orientations and their corresponding selected Fourier transformation spots as insets. **h** Superposition image of lithium oxide. **i** A sodium particle grown along [11$\bar{2}$] direction. **j** AC-HRTEM of the particle, (**k**) partial enlarged view and simulated AC-HRTEM of sodium as inset. **l** EFTEM of a sodium particle. **m** Sodium oxide on the edge and its Fourier transformation spots inset. **n, o** Sodium oxide compounds, sodium, and their corresponding selected Fourier transformation spots inset, respectively. **p** Superposition image of different compounds. **f–h, n–p** These images are pseudo-colored.

analysis further proves that lithium is in the form of single crystal while surface $Li_2O$ is polycrystalline.

Similarly, Fig. 4i–p depicts the condition of sodium particles (dose-rate ~850 e Å$^{-2}$ s$^{-1}$). The single-crystalline feature of sodium particle can be observed, covered by surface oxide compounds. Besides the AC-HRTEM analysis, dark field TEM experiments were carried out and the results (Supplementary Fig. 8) agree with high magnification observations. Moreover, the thickness of lithium and sodium particles was measured taking

advantage of the EFTEM technique. The average thickness is normally <70 nm (Supplementary Fig. 9). The difference in sodium particles is that the oxide layers (in Fig. 4m) are thinner than those of lithium particles (in Fig. 4e), as the AC-HRTEM observation of sodium particles was carried out after they have been placed in TEM vacuum chamber for only several minutes, shorter than the case of lithium particles. This indicates that the formation of oxide layer could be dependent of the time the particles exposed inside the vacuum chamber. Further

comparison experiments confirm such deduction. As detailed in Supplementary Fig. 10, the thickness of oxide layers increased with the time exposed in the vacuum chamber. Meanwhile, in SAED experiments, the enhancement of the signal of $Li_2O$ polycrystalline rings versus time also supports this conclusion. To be noted that electron beam was blanked during the oxide layer thickening process (Supplementary Fig. 10), so that the possibility of beam-induced oxidation could be ruled out.

Although alkali metal is easily to be oxidized, it is puzzling oxidation could occur in the TEM vacuum chamber ($10^{-5}$ Pa). In the next step, the oxidation is investigated. For the alkali metal grown from the carbonate, one may doubt that the oxygen is originated from carbonate itself. However, the comparison experiments performed on lithium fluorides indicate that oxidation could occur even there was no oxygen source (Supplementary Fig. 5). This can be further proved by detailed energy dispersive spectroscopy (EDS) and electron energy loss spectroscopy (EELS) analysis (Supplementary Figs. 11–13). An in situ comparison EDS detection has revealed that there was almost no oxygen signal before the formation of lithium particles from LiF whereas relatively large amount of oxygen was detected at the end (Supplementary Fig. 11). Such difference rules out the oxygen source from the starting material as well as possible environmental oxygen adsorption on its surface. EDS and EELS mapping (Supplementary Figs. 12, 13) confirms our previous AC-HRTEM results (Fig. 4) that oxides were distributed on the surface of the particles. To further rule out the possible environmental oxygen adsorption on the carbon membrane of TEM grids, comparison experiments were carried out on pure gold and nickel-metal grids without any supporting membranes. It turned out that oxidation still occurred on the initially formed lithium particles (Supplementary Fig. 14). On this basis, we attribute the oxidation to the possible existence of trace oxygen in the high vacuum chamber, and interestingly alkali metals could react with the trace oxygen efficiently and result in oxygen segregation as oxides. Moreover, cooling experiments were also performed. Compared with the room temperature condition, alkali metal particles almost could not be generated under cryogenic condition at −178 °C (Supplementary Movies 7, 8). Lowering the temperature may increase the energy barrier for the alkali metal formation so that the formation of alkali metal particles cannot be observed.

**Ambiguities in lithium metal growth on garnet-type solid electrolytes of lithium-ion batteries**. As one of the important candidates of the next generation solid electrolyte, $Li_7La_3Zr_2O_{12}$ (LLZO) and its derivates have been widely investigated because of their relatively high ionic conductivity and good chemical stability against lithium metal anode[24,25]. It has been reported by different research groups that lithium metal could be grown from LLZO under the electron beam irradiation[26,27]. Here we employed our method on LLZO and our observations indicate lithium metal growth on LLZO could be an illusion.

At low magnification, growth of lithium particles from the surface of LLZO could be observed, as shown in Fig. 5a (indicated by red arrows). Such phenomenon is similar with the observations in Fig. 1, and looks in agreement with previous reports using scanning electron microscopy (SEM)[26,27]. However, a detailed analysis on the thinner edge of the LLZO particle shows that the edge is lithium carbonate instead of LLZO. The SAED pattern in Fig. 5b is indexed to be lithium carbonate, corresponding to the selected area in Fig. 5a. Therefore, we conclude that lithium metal is grown from the lithium carbonate, which is the well-known surface contamination of LLZO. By comparison, we selected a clean LLZO particle and the phenomenon of lithium metal

growth could not be observed (Fig. 5c). SAED on the edge of the particle suggests pure phase of LLZO, as shown in Fig. 5d. The only effect of electron beam bombardment is that the surface of LLZO turns into amorphous layer (outside the orange dotted line in Fig. 5c). Hence, compared with previous reports, our high-resolution observations indicate that lithium metal could not be grown on pure LLZO phase and the true origin may come from lithium carbonate contamination, which is easily formed on the garnet surface[28]. In contrast to low-resolution analysis methods such as using SEM, the combination of low dose-high-resolution imaging, electron diffraction in TEM as well as the EELS elemental determination is the key to comprehensive analysis of lithium growth phenomenon.

**Contact property of lithium metal**. The research on lithium-lithium contact as well as lithium-lithium oxide contact is difficult to carry out in a conventional way. Another advantage of our method we demonstrate here is that the physical contact property of lithium metal and lithium oxide could be studied. Wettability of liquid lithium has been studied[29] whereas the property of solid lithium contact is less known. On the other hand, contact between lithium metal and lithium oxide is of interest as lithium oxide is an important inorganic component of solid electrolyte interphase (SEI) on the surface of lithium dendrite in batteries[7].

The experiment setup is shown in Fig. 6a. Utilizing a scanning tunneling microscopy (STM) tip inside the microscope, lithium whisker could be produced by attaching the tip to the lithium particle and then pulling away. The black circle represents the area irradiated by electron beam for the generation of lithium. The in situ formation process of lithium whisker is shown in Supplementary Movie 9. Continuously stretching of the whisker would break it and fresh lithium tips were exposed (Fig. 6b and Supplementary Movie 10). Therefore, the contact property could be studied using these lithium tips. Firstly, Li–Li contact is demonstrated in Fig. 6c and Supplementary Movie 11. Pushing two Li tips to contact with each other right after the breakdown, the two tips can recombine together into one whisker again. On contrast, the $Li_2O$–$Li_2O$ contact is demonstrated in Fig. 6d and Supplementary Movie 12. Waiting for several minutes after the whisker breakdown, the surface of Li tips was covered by oxide layers. When the two tips got in touch, they could not recombine. Further pushing of two tips caused deformation on both sides. Finally, the Li–$Li_2O$ contact is demonstrated in Fig. 6e and Supplementary Movie 13. When a fresh Li tip contacted with an oxidized tip, they could combine together. The corresponding schematic diagrams for all these cases are shown in Fig. 6f–j.

To better understand the above contact properties, first-principle calculations were performed. The interfacial work of adhesion $W_{ad}$ between two surfaces is chosen as the criterion to evaluate the interaction between two surfaces. Figure 6k–n shows the relaxed interface structures of Li(110)-Li(110), $Li_2O$(001)-$Li_2O$(001), and two types of Li(001)-$Li_2O$(001) interfaces, respectively. The interfacial work of adhesion was calculated to be −1.01 J m$^{-2}$ for Li(110)-Li(110), and −1.21 J m$^{-2}$ for Li(001)-$Li_2O$(001) with Li–Li contact at the interface, and −6.43 J m$^{-2}$ for Li(001)-$Li_2O$(001) with Li–O contact at the interface. The negative values of $W_{ad}$ suggest the combination of the two surfaces is energetically favorable, in good agreement with the experimental results. However, $W_{ad}$ was calculated to be −7.92 J m$^{-2}$ for $Li_2O$(001)-$Li_2O$(001), which is in contradiction with the experiment that oxide layers could not be combined together. It is speculated that the discrepancy is related to the polycrystalline feature of the lithium oxide layers. Random orientation of lithium oxides and grain boundaries may increase the interfacial work of adhesion so that the contact becomes unfavorable. The good

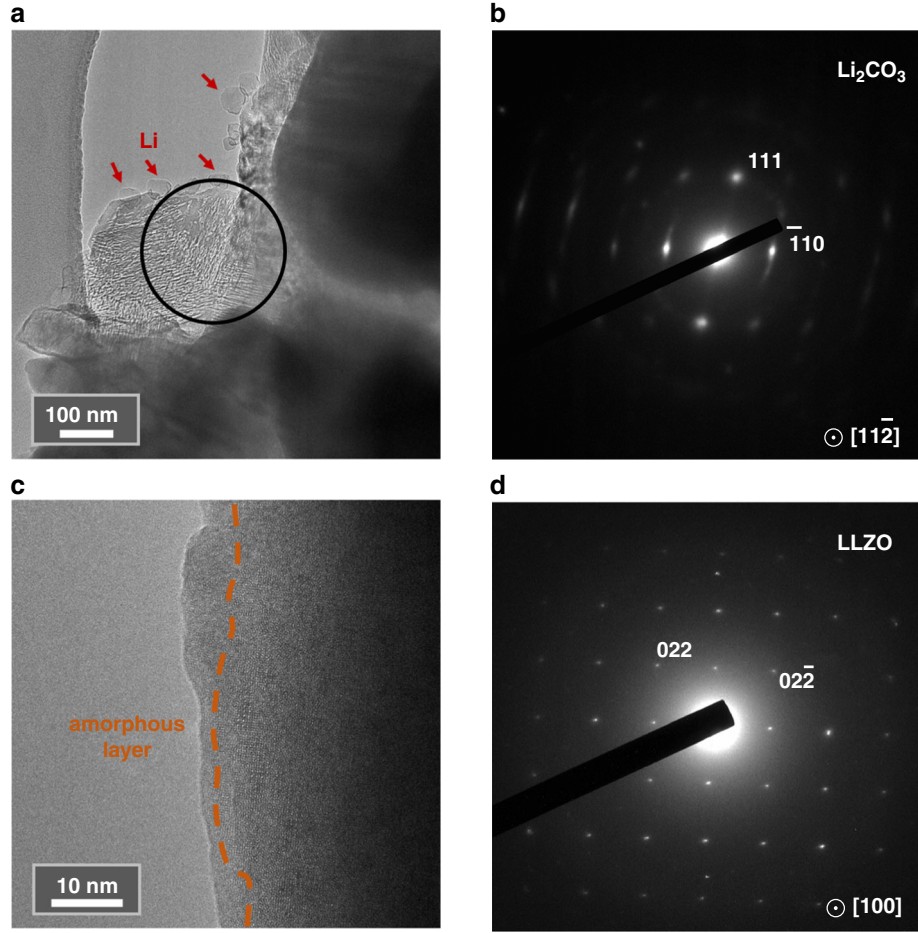

**Fig. 5 Lithium growth from the surface contamination layer of LLZO. a** Lithium particles (indicated by red arrows) grown from lithium carbonate contamination layer. Black circle represents the selected area for SAED in (**b**). **b** the SAED pattern is indexed to be $Li_2CO_3$. **c** No lithium particle growth can be observed on pure surface of LLZO upon electron irradiation. Orange dotted lines mark the amorphous layer caused by irradiation. **d** SAED of the region in (**c**) is indexed to be LLZO.

contact between lithium metal and others could be closely related with the liquid-like fluidity of solid lithium as shown in these experiments. Our results on the basic contact property of lithium metal and surface passivation oxide layer provide support for better understanding of lithium dendrite growth and SEI related issues in batteries. For example, the phenomenon of "dead lithium" finds support from the bad contact property between surface passivation layers.

**Mechanism of the alkali metal formation**. Finally, combining with first-principle calculations, the formation mechanism of alkali metal particles is discussed. It is clear that the reaction is initiated by the electron beam, which deposits energy to the materials when it is incident to them. The interaction between high energy particles and materials[30–33] can result in thermal heating[30], knock-on damage[31], radiolysis damage[32], and charging effect[32,33]. In terms of thermal decomposition, in situ heating experiment of LiF (Supplementary Fig. 15) indicates that heating effect shouldn't be the main driving force for the formation of alkali particles, as alkali metal particles did not show up just by heating. Furthermore, first-principle calculations (Supplementary Fig. 16) of the thermal decompositions for carbonates and fluorides under the ambient and vacuum conditions were performed, and pure alkali metals could not be the products thermodynamically, confirming that thermal decomposition should

not be the predominant reaction mechanism. Hence, the formation of alkali metal is attributed to the radiolysis and charging effect. Irradiation of alkali salts can promote the formation of Frenkel defects in the crystal, and mobile state F* centers can recombine with surface terrace edge and initiate emission of alkali atoms[21,34]. These alkali atoms can aggregate on the surface of alkali salts and form alkali metals as a result. While the growth of alkali metal particles and whiskers could be the competition between knock-on damage and diffusion flux.

## Discussion
We propose a simple and versatile method to form alkali metals in situ inside TEM directly. With this method, we have visualized the growth of alkali metals at atomic spatial resolution and millisecond temporal resolution. Oxidation of the alkali metals was observed, and the distribution and formation of surface oxide component was investigated. Throughout the text, this method shows its great compatibility in different experiments, and its applications in other in situ experiments are also predictable. As practical applications, we clarify the ambiguities in lithium metal growth on garnet-type solid electrolytes for lithium-metal batteries. On the other hand, we demonstrate a direct way to study the contact property of lithium metal as well as its surface passivation oxide layer.

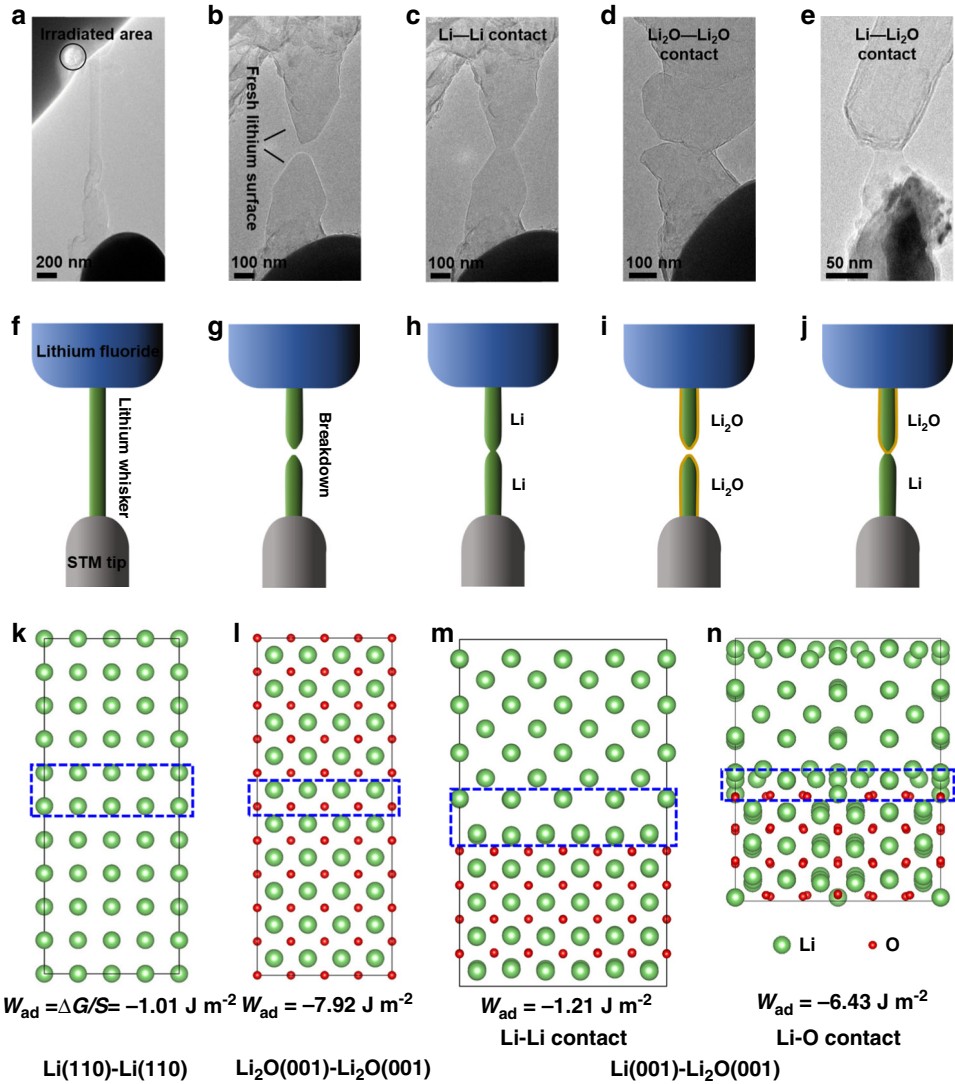

**Fig. 6 Contact property of lithium metal and surface oxide layer. a** Formation of lithium whisker with the help of an STM tip. The black circle represents the area where initial lithium particles were formed. **b** Breakdown of the lithium whisker, and two fresh lithium tips are obtained. **c** Li–Li contact. **d** $Li_2O$-$Li_2O$ contact. **e** Li–$Li_2O$ contact. **f–j** Corresponding schematics of (**a–e**). **k** Li(110)-Li(110) interface model, **l** $Li_2O$(001)-$Li_2O$(001) interface model, **m** Li(001)-$Li_2O$ (001) with Li–Li contact at the interface, **n** Li(001)-$Li_2O$(001) with Li–O contact at the interface. The interface regions are labeled by blue dotted boxes.

## Methods

**Materials**. For the electron microscopy study, pure lithium carbonate ($Li_2CO_3$, 99.998%, from Aladdin), sodium carbonate ($Na_2CO_3$, 99.8%, from Tansoole), lithium fluoride (LiF, 99.99%, from Aladdin) were used.

**X-ray diffraction and electron microscopy**. Powder X-ray diffraction patterns were obtained using a Bruker AXS D8 Advance diffractometer with a Cu Kα source ($\lambda_{Cu\ K\alpha}$ = 1.54 Å).

TEM experiments were performed on JEOL JEM-2100 Plus (200 kV) and double aberration-corrected JEOL GrandArm (300 kV). To be specific, HRTEM images in Figs. 3–4, EFTEM images and EELS were obtained on JEOL GrandArm. The Gatan Oneview IS camera enabled fast in situ data acquisition. Other TEM results were obtained on JEOL JEM-2100 Plus. Cryogenic experiments were carried out with Gatan double tilt cooling holder (Model 636) which can sustain a low-temperature environment at −178 °C. In situ heating experiments were carried out with Protochips (Fusion 350) in situ heating holder. Lithium whisker contact experiments were carried out with an in situ STM tip holder (PicoFemto). AC-HRTEM images of lithium and sodium particles were collected under the negative spherical aberration ($C_S$) imaging (NCSI) condition. Average background subtracted filtering was carried out based on the script from D. R. G. Mitchell and method by Kilaas[35].

## Data availability

The data support the findings of this manuscript are available from the corresponding author upon reasonable request.

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

## Acknowledgements

This work was supported by the National Key Research and Development Program of China (2019YFA0210600), the National Natural Science Foundation of China (21805184, 21805185), the Natural Science Foundation of Shanghai (18ZR1425200), and startup funding from ShanghaiTech University. The TEM characterization was supported by the Center for High-resolution Electron Microscopy (ChEM), and the XRD (Dr. N. Yu) measurement was supported by the SPST Analytical Center at ShanghaiTech University. The theoretical simulation is supported by HPC Platform of ShanghaiTech University.

## Author contributions

C.L. and Y.Y. conceived the idea and designed the experiments. C.L. and Y.Y. conducted TEM characterization. X.Z. conducted first principle calculations. S.X. conducted XRD characterization and sample preparation. Z.W. performed EELS analysis. J.W. completed schematic drawing. B.Y. assisted with image filtering process. X.L. helped with videos compression. C.L. and Weiyan L. performed in situ heating experiment. Wei L. contributed to key discussions of the results and manuscript writing. C.L., X.Z., Wei L., and Y.Y. interpreted the results. C.L., X.Z., Z.W., Wei L., and Y.Y. co-wrote the paper. All authors discussed the results and commented on the manuscript.

## Competing interests

The authors declare no competing interests.
