## [Peer Review File · Nature Communications]

Reviewers' comments:

Reviewer #1 (Remarks to the Author):

Comments on:

"Imaging the In situ Production of Alkali Metals at High Spatiotemporal Resolution" by C. Liang et al.

The authors describe the formation of pure alkali metals during electron irradiation of salts containing these elements. They attribute the metal formation to a reduction process and put forward a hypothesis how it may occur. Their observations are made in situ and at variable temperature. It is indeed true that the scientific community struggles to directly investigate the structure of alkali metals because of their reactivity if exposed to air. This work describes an elegant path how to make them by electron irradiation if needed. Obviously dose and dose rate considerations are important in this context since the electron beam is part of the experiment. As such, it is appropriate to publish this work.

However, one must put this research into the broader context of what is known and was done in the past. For example, a beam induced reduction of metals or beam induced crystal growth are studied since long (e.g. E.A. Stach et al, Nano Letters 3, 2003, 867) with a comparably low spatial and temporal resolution. In fact it is generally accepted that the electron beam drives chemical reactions. However, details of chemical processes are commonly obscured by the lack of temporal and spatial resolution in electron microscopy. Electronic excitations typically occur on a time scale of femtoseconds, atom displacements at phonon frequencies of picoseconds and even atom diffusion is very fast compared to the time scale of milliseconds, which is achieved here. Factually, the needed temporal resolution to study chemical reactions is only emerging and a high spatial resolution is commonly not achievable because of dose limitations. Thus, the general claim that dynamic processes are tracked only relates to the observation that nanoparticles of alkali metals assume a particular shape during growth, which is not quite unexpected. The claim seems overemphasized. In addition, the Fig.5, "Mechanism diagram of the reduction of alkali metals", seems misleading since the electron beam does not only pass the sample at a distance in the vacuum. Instead the beam electrons penetrate the entire picture like a shower across the sample and the vacuum. As a result charge accumulations are ill defined and charge repulsion or attraction may not be directional at all, which questions their explanation.

Consequently, the relevance of this observation to the scientific community is debatable and the question where to publish is appropriate. 
There are minor aspects that need consideration

The word "production" relates to a manufacturing process, which is not what the authors do. They describe a formation process.

Oxydation can occur via beam-induced surface diffusion or by residual gases in the column. It is not clear what is dominant here.

Nowadays a Fourier Transform is always "fast". There is no need to call it FFT, which is an obsolete expression from the time when computers were slow.

"Reciprocal Space Length" is an unnecessary new expression, which is typically called "scattering vector" for example.

A growth process is by definition dynamic, there is no need to additionally add the word "dynamic".

The authors claim on page 3 that this is a one step reaction. However, even their reaction pathways in the SI are made from several steps in the most simple approximation.

Reviewer #2 (Remarks to the Author):

In this manuscript, the authors developed a method to produce the alkali metals in situ inside the TEM, and observed the atomic structure, dynamic growth and oxidation of alkali metals. The reported in situ alkali metal preparation technique is novel and certainly contains merit, however the significance of the results cannot be fully justified for publication in Nature communications. I suggest the authors to improve their manuscript by considering the following comments/suggestions.

1. The authors may need to clarify the new insights obtained in this work compared to previous results, as alkali metal structures are well known and have already been observed, eg. [Science 358, 506-510, 2017; Nat. Nanotechnol. (2019) doi:10.1038/s41565-019-0558-z]. Also, lithium fiber growth was studied by a recent paper [Nat. Nanotechnol. (2019) doi:10.1038/s41565-019-0558-z]. The authors could make a comparison of the Li growth behavior between this work and the Nat. Nanotechnol. paper, which may help to dig deeper into the growth mechanism. The authors may also want to emphasize the significance of their results, e.g., are they of any relevance for practical applications?
2. Detailed mechanisms for the alkali metal growth process are lacking. For example, is the growth process diffusion-mediated? If yes, is it mainly composed of surface- or bulk-diffusion?
3. Although the mechanism for alkali metal production is mainly radiolysis and charging effect, knock-on effect cannot be ignored during the subsequent growth period, especially for light elements such as Li. In my opinion, the growth process is very likely a competition between irradiation damage and diffusion flux. If the diffusion flux is approximately constant, the knock-on effect gets more and more severe as the metal foil grows, which slows down the growth rate, as observed in Fig. 1. Oxidation may further reduce the growth rate by either confinement (bulk diffusion dominated) or reduction in diffusion flux (surface diffusion dominated).
4. After long-time irradiation, lithium may be depleted while only a lithium oxide shell is left. Thus, the oxide layer thickening mechanism observed in this work could be different from those oxidation process we normally observe.
5. Can the authors provide any evidence to prove the produced metals are in the form of a foil (as claimed in the manuscript) rather than a nanoparticle? This reflects the growth mode, i.e., 2D- vs 3D-growth.
6. The claimed "layer-by-layer" growth mechanism is not straightforward in Fig. 3.
7. In Fig. 4f-h, can the authors specify how is the mapping performed? In Fig. 4d and 4l, are there other elements, such as oxygen, besides alkali metals? Sodium oxide (Fig. 4m) appears differently compared to lithium oxide (Fig. 4e). Have the authors investigated whether sodium oxide is single crystalline or polycrystalline?
8. The vacuum inside the TEM (10^{-5} Pa) is not quite qualified for ultrahigh vacuum as mentioned in the abstract.

Reviewer #3 (Remarks to the Author):

In their manuscript entitled "Imaging the In situ Production of Alkali Metals at High Spatiotemporal Resolution" C. Liang, X. Zhang, S. Xia, Z. Wang, J. Wu, B. Yuan, X. Luo, W. Liu, W. Liu and Y. Yu report on detailed investigation of leaching and crystallization of alkali metals - lithium and sodium - from their corresponding salts (fluorides and carbonates) upon exposure to electron beam. Crystallization of alkali metals followed by formation of thin oxide layer was observed even under

ultra-high vacuum conditions and was extensively studied by various electron microscopy techniques (diffraction, atomic resolution imaging, EELS and EDX). Various factors, such as low and moderate temperatures, type of TEM grid were considered to elucidate the mechanism of reduction of alkali metals by electron beam.

This is a well-rounded study of the chemical phenomena occurring under very special conditions (electron beam and ultra-high vacuum). While this is an interesting study, which may contribute to our understanding of electron beam interactions with matter, I do not see how this study "may lead to a more complete understanding of the structure and property 216 of Group I elements," as claimed by Authors. Overall, I can recommend this paper to be published in Nature Communications, but urge Authors to eliminate unsupported claims from the manuscript.

Response to Reviewer's Comments for Manuscript NCOMMS-19-33196

Response to Reviewer 1

Reviewer #1:

Comments:

The authors describe the formation of pure alkali metals during electron irradiation of salts containing these elements. They attribute the metal formation to a reduction process and put forward a hypothesis how it may occur. Their observations are made in situ and at variable temperature. It is indeed true that the scientific community struggles to directly investigate the structure of alkali metals because of their reactivity if exposed to air. This work describes an elegant path how to make them by electron irradiation if needed. Obviously dose and dose rate considerations are important in this context since the electron beam is part of the experiment. As such, it is appropriate to publish this work.

However, one must put this research into the broader context of what is known and was done in the past. For example, a beam induced reduction of metals or beam induced crystal growth are studied since long (e.g. E.A. Stach et al, Nano Letters 3, 2003, 867) with a comparably low spatial and temporal resolution. In fact it is generally accepted that the electron beam drives chemical reactions. However, details of chemical processes are commonly obscured by the lack of temporal and spatial resolution in electron microscopy. Electronic excitations typically occur on a time scale of femtoseconds, atom displacements at phonon frequencies of picoseconds and even atom diffusion is very fast compared to the time scale of milliseconds, which is achieved here. Factually, the needed temporal resolution to study chemical reactions is only emerging and a high spatial resolution is commonly not achievable because of dose limitations. Thus, the general claim that dynamic processes are tracked only

relates to the observation that nanoparticles of alkali metals assume a particular shape during growth, which is not quite unexpected. The claim seems overemphasized.

In addition, the Fig.5, “Mechanism diagram of the reduction of alkali metals”, seems misleading since the electron beam does not only pass the sample at a distance in the vacuum. Instead the beam electrons penetrate the entire picture like a shower across the sample and the vacuum. As a result charge accumulations are ill defined and charge repulsion or attraction may not be directional at all, which questions their explanation.

Consequently, the relevance of this observation to the scientific community is debatable and the question where to publish is appropriate.

Author response:

We highly appreciate the reviewer’s comments on our manuscript, and we have improved the manuscript accordingly.

As suggested, we have removed those misleading and unsupported claims. On one hand, we agree with the reviewer that temporal resolution achieved here is not enough for the observation of femtosecond chemical reactions and we have removed the misleading title/descriptions. On this basis, we focus on the in-situ growth process of the lithium particles and the growth kinetics has been investigated in detail. On the other hand, we removed the original Fig.5 and its related unsupported claims. Discussion of the mechanism of the alkali metal formation has been made combining our results with previous literatures. More importantly, following the reviewer’s comments, we have referred to previous works related to in-situ observation of crystal growth, and we have done substantial amount of additional experiments to show the impact of our methods demonstrated here. Major changes are listed below:

- (1) In the revised manuscript, the title has been changed to “Unravelling the room-temperature atomic structure and growth kinetics of lithium metal”.
- (2) Fig. 3 has been updated with a detailed analysis of the growth of lithium

particles. Besides, we have demonstrated the growth of lithium dendritic whiskers. Quantitative investigation of the whisker growth kinetics here has been compared with the electrical current-driven growth of lithium dendrites in lithium ion batteries.

Fig. 3 In situ growth of lithium particle at high spatiotemporal resolution. a-c

Growth along $[10\bar{1}]$ direction. **a** A $[111]$ -oriented lithium particle grown from lithium fluoride. Inset is the corresponding Fourier transformation pattern. Red single solid lines represent the edge of this particle. Image filtering was applied to enhance the signal to noise ratio. Dose-rate is $\sim 1000 \text{ e}\text{\AA}^{-2}\text{s}^{-1}$. **b** Edge of the particle at this stage are tracked by red dotted lines. The middle area between solid lines and dotted lines represents the additive grown portion. **c** Edge of the particle at this stage are tracked by red double solid lines. The middle area between solid lines and dotted lines represents the additive grown portion. **d-f** Growth along $[\bar{1}10]$ direction. **d** Red single solid lines represent the edge of this particle. $(2\bar{1}\bar{1})$ surface is exposed and the corresponding Fourier transformation spot is marked by red circle. **e** Edge of the

particle at this stage are tracked by red dotted lines. **f** Edge of the particle at this stage are tracked by red double solid lines. **g** Growth length versus time. Arrows represent starts of new growing processes. **h** Growth rate versus time. Arrows mark the growing steps as in (g), respectively. **i-j** Images of two types of lithium whisker growth, Type I in (i) and Type II in (j), respectively. **k-l** Growth length (k) and growth rate (l) versus time.

(3) Fig. 5 has been removed. We have updated a new Fig.5, demonstrating an immediate advantage of our study is that, ambiguities in lithium metal growth on garnet-type solid electrolytes of lithium ion batteries could be solved. We employed our method on $\text{Li}_7\text{La}_3\text{Zr}_2\text{O}_{12}$ (LLZO) and our observations indicated that previous reported lithium metal growth on LLZO could be an artifact and the true origin came from surface contaminated lithium carbonates.

Fig.5 Lithium growth from the surface contamination layer of LLZO. **a** Lithium particles (indicated by red arrows) growth from lithium carbonate contamination layer. Black circle represents the selected area for SAED in (b). **b** the SAED pattern is indexed to be Li_2CO_3 . **c** No lithium particle growth can be observed on pure surface of LLZO upon electron irradiation. Orange dotted lines mark the amorphous layer caused by irradiation. **d** SAED of the region in (c) is indexed to be LLZO.

(4) Fig.6, i.e., a new figure has been added, demonstrating an application of our

method is that, the physical contact property of lithium metal and surface passivation oxide layer could be studied. By introducing an AFM tip inside the TEM, we were able to guide the growth of lithium whisker and study the contact property of Li-Li, Li₂O-Li₂O, and Li-Li₂O, respectively. These results may provide better understanding of lithium dendrite growth and solid electrolyte interphase (SEI) related issues in batteries.

Fig.6 Contact property of lithium metal and surface oxide layer. **a** Formation of lithium whisker with the help of an AFM tip. The black circle represents the area where initial lithium particles were formed. **b** Breakdown of the lithium whisker, and two fresh lithium tips are obtained. **c** Li-Li contact. **d** Li₂O-Li₂O contact. **e** Li-Li₂O contact. **f-j** Corresponding schematics of **a-e**. **k** Li(110)-Li(110) interface model, **l** Li₂O(001)-Li₂O(001) interface model, **m** Li(001)-Li₂O(001) with Li-Li contact at the interface, **n** Li(001)-Li₂O(001) with Li-O contact at the interface. The interface regions are labeled by blue dotted box.

With these improvements, we believe our in-situ method to observe the growth of alkali metals show different features and advantages compared with previous TEM studies of crystal growth. Atomic resolution imaging of beam-sensitive alkali metals

at room temperature and the research on their basic structure and properties may have positive influence on their applications such as lithium ion batteries.

Our responses to the minor comments are provided below.

There are minor aspects that need consideration

The word “production” relates to a manufacturing process, which is not what the authors do. They describe a formation process.

Author response:

We have changed “production” to “formation” as suggested.

Oxydation can occur via beam-induced surface diffusion or by residual gases in the column. It is not clear what is dominant here.

Author response:

We propose the residual gases in the column may be the dominant reason. The conclusion comes from the comparison experiments between the lithium carbonate and lithium fluoride. For the case of lithium growth from lithium fluoride, there’s no oxygen element in the alkali salts and unlikely to provide adequate source of oxygen. Besides, even without beam irradiation, oxidation could still be observed for the samples placed in the TEM column with beam-off (Supplementary Fig.5 & 10).

In addition, the comparison experiments of EDS (Fig.S11), change of TEM grids (Fig.S14), and heating (Fig.S15) rule out the possible absorbed oxygenic gas on the sample. Moreover, comparison experiments outside the TEM were also performed. For this purpose, the lithium fluoride sample was placed into the argon filled glove box and heated to 150 °C to desorb the oxygen/vapor/carbon dioxide as much as possible. The heating process was sustained for two hours and only slightly rise of oxygen and vapor has been observed (from ~0.5ppm to ~9.5ppm) during this process. This ex-situ experiments are in agreement with the in-situ TEM-EDS (Fig.S11), demonstrating that surface absorbed oxygen may not be the major oxygen source.

Author action:

We have added the discussions into the main text (Page 10).

Nowadays a Fourier Transform is always “fast”. There is no need to call it FFT, which is an obsolete expression from the time when computers were slow. “Reciprocal Space Length” is an unnecessary new expression, which is typically called “scattering vector” for example.

Author response:

We have changed FFT to Fourier transformation.

A growth process is by definition dynamic, there is no need to additionally add the word “dynamic”.

Author response:

We have deleted the word “dynamic”.

The authors claim on page 3 that this is a one step reaction. However, even their reaction pathways in the SI are made from several steps in the most simple approximation.

Author response:

We are sorry that our original descriptions mislead the reviewer. The possible reaction pathways provided in the main text of SI as well as in Fig.S16 listed both one-step and two-step reactions for comparison. For two-step reactions, alkali metal oxides are the intermediate products. In the manuscript, our observations supported the one-step reaction as we observed alkali metals first and then oxidation occurred.

Author action:

We have made the description clear in the main text (Page 14) and SI.

Response to Reviewer 2

Reviewer #2:

Comments:

In this manuscript, the authors developed a method to produce the alkali metals in situ inside the TEM, and observed the atomic structure, dynamic growth and oxidation of alkali metals. The reported in situ alkali metal preparation technique is novel and certainly contains merit, however the significance of the results cannot be fully justified for publication in Nature communications. I suggest the authors to improve their manuscript by considering the following comments/suggestions.

Author response:

We highly appreciate the reviewer's comments and our responses to the specific comments are provided below.

1. The authors may need to clarify the new insights obtained in this work compared to previous results, as alkali metal structures are well known and have already been observed, eg. [Science 358, 506-510, 2017; Nat. Nanotechnol. (2019) doi:10.1038/s41565-019-0558-z]. Also, lithium fiber growth was studied by a recent paper [Nat. Nanotechnol. (2019) doi:10.1038/s41565-019-0558-z]. The authors could make a comparison of the Li growth behavior between this work and the Nat. Nanotechnol. paper, which may help to dig deeper into the growth mechanism. The authors may also want to emphasize the significance of their results, e.g., are they of any relevance for practical applications?

Author response:

We highly appreciate the nice suggestion of comparing our results with previous works, especially for the recent Nat. Nanotechnol. (14, 1042-1047, 2019) publication, which we haven't seen at the time the manuscript was prepared. Making comparison

between the lithium growth behavior driven by electron beam irradiation (this work) and by electrical current (Nat. Nanotechnol.) do help for the improvement of our manuscript and the understanding may be strengthened.

In the revised manuscript, two major changes related to this concern are listed below:

- (1) The title has been changed to “Unravelling the room-temperature atomic structure and growth kinetics of lithium metal”.
- (2) Fig. 3 has been updated with a detailed analysis of the growth of lithium particles. Besides, we have demonstrated that with our method we could also observe the growth of lithium dendritic whiskers. Quantitative investigation of the whisker growth kinetics here has been compared with the electrical current-driven growth of lithium dendrites in previous works, including Ref.7: Science, 358, 506-510, (2017), Ref.19: Nat. Nanotechnol., 14, 1042-1047, (2019), and Ref.20: Nano Energy, 32, 271-279, (2017).

Fig. 3 In situ growth of lithium particle at high spatiotemporal resolution. a-c

Growth along $[10\bar{1}]$ direction. **a** A $[111]$ -oriented lithium particle grown from lithium fluoride. Inset is the corresponding Fourier transformation pattern. Red single solid lines represent the edge of this particle. Image filtering was applied to enhance the signal to noise ratio. Dose-rate is $\sim 1000 \text{ e}\text{\AA}^{-2}\text{s}^{-1}$. **b** Edge of the particle at this stage are tracked by red dotted lines. The middle area between solid lines and dotted lines represents the additive grown portion. **c** Edge of the particle at this stage are tracked by red double solid lines. The middle area between solid lines and dotted lines represents the additive grown portion. **d-f** Growth along $[\bar{1}10]$ direction. **d** Red single

solid lines represent the edge of this particle. $(2\bar{1}\bar{1})$ surface is exposed and the corresponding Fourier transformation spot is marked by red circle. **e** Edge of the particle at this stage are tracked by red dotted lines. **f** Edge of the particle at this stage are tracked by red double solid lines. **g** Growth length versus time. Arrows represent starts of new growing processes. **h** Growth rate versus time. Arrows mark the growing steps as in (g), respectively. **i-j** Images of two types of lithium whisker growth, Type I in (i) and Type II in (j), respectively. **k-l** Growth length (k) and growth rate (l) versus time.

Detailed discussions of the comparisons are provided below:

Figures 3i-l show the growth and kinetics of lithium whisker from an as-synthesized lithium particle (defined as Type I, Supplementary Video 5) and from the root lithium fluoride materials (defined as Type II, Supplementary Video 6), respectively. Detailed descriptions have been provided in the revised manuscript. According to our observations, the growth behavior of lithium in vacuum driven by electron beam irradiation is in general similar with the situation in liquid electrolyte and in environmental gas (Ref.19, 20) driven by electrical current.

To be specific, the morphology of lithium particle (Fig. 1) and lithium whisker (Fig. 3i and j) in our case is similar with the initial lithium particle and sequent grown lithium whisker in Ref.19, respectively. For the growth kinetics, it is also similar by comparing Fig. 3h in this work with Fig. 2h in Ref.19. The whisker growth in Fig. 2h of Ref.19 shows an accelerated growth process (even under increasing stress) which is similar to the Type II growth in our case. Meanwhile, the growth kinetics we observed is in agreement with the in-situ observations in liquid electrolyte (Ref.20). Furthermore, the cross section of our lithium whiskers (Fig.S4) is similar with that prepared by electrochemical deposition in Ref.7.

On this basis, it seems that different environments (CO_2 , N_2 , vacuum, and liquid) have little effect on the growth kinetics of lithium metal, whereas the morphology is related to the surface passivation layers, which is sensitive to the surrounding environment. For example, in Ref. 19, initial growth of lithium particle and subsequent growth of lithium whisker was observed in CO_2 environment (1 Pa), while growth of lithium particle was dominated in N_2 environment (1 Pa). In our case without gas environments and elastic constraints, both particles and whiskers have been observed. And whiskers could be grown either from as-formed lithium particles or the root lithium salts. It is assumed that the 1D- or 3D-growth mode is on one hand related to the confinement from surface passivation layers, and on the other hand related to the diffusion flux which is dependent of the strength of the driving force

(electron beam irradiation or electrical current).

Consequently, apart from the methods demonstrated in Ref.7, 19, and 20, our method may provide an alternative way to study lithium metal-related structure problems as high spatiotemporal resolution is a major advantage here.

For the second concern raised by the reviewer is the significance of our results. We also take the nice suggestion from the reviewer and have done substantial amount of additional experiments to further demonstrate the impact of our observations. Two major changes related to this concern are listed below:

- (1) The original Fig. 5 has been removed. We have updated a new Fig.5, demonstrating an immediate advantage of our study is that, ambiguities in lithium metal growth on garnet-type solid electrolytes of lithium ion batteries could be solved. We employed our method on $\text{Li}_7\text{La}_3\text{Zr}_2\text{O}_{12}$ (LLZO) and our observations indicated that previous reported lithium metal growth on LLZO could be an artifact and the true origin came from surface contaminated lithium carbonates.

Fig.5 Lithium growth from the surface contamination layer of LLZO. a Lithium particles (indicated by red arrows) growth from lithium carbonate contamination layer.

Black circle represents the selected area for SAED in (b). **b** the SAED pattern is indexed to be Li_2CO_3 . **c** No lithium particle growth can be observed on pure surface of LLZO upon electron irradiation. Orange dotted lines mark the amorphous layer caused by irradiation. **d** SAED of the region in (c) is indexed to be LLZO.

(2) Fig.6, i.e., a new figure has been added, demonstrating an application of our method is that, the physical contact property of lithium metal and surface passivation oxide layer could be studied. By introducing an AFM tip inside the TEM, we were able to guide the growth of lithium whisker and study the contact property of Li-Li, Li_2O - Li_2O , and Li- Li_2O , respectively. These results may provide better understanding of lithium dendrite growth and solid electrolyte interphase (SEI) related issues in batteries.

Fig.6 Contact property of lithium metal and surface oxide layer. **a** Formation of lithium whisker with the help of an AFM tip. The black circle represents the area where initial lithium particles were formed. **b** Breakdown of the lithium whisker, and two fresh lithium tips are obtained. **c** Li-Li contact. **d** Li_2O - Li_2O contact. **e** Li- Li_2O contact. **f-j** Corresponding schematics of **a-e**. **k** Li(110)-Li(110) interface model, **l** Li_2O (001)- Li_2O (001) interface model, **m** Li(001)- Li_2O (001) with Li-Li contact at the interface, **n** Li(001)- Li_2O (001) with Li-O contact at the interface. The interface

regions are labeled by blue dotted box.

2. Detailed mechanisms for the alkali metal growth process are lacking. For example, is the growth process diffusion-mediated? If yes, is it mainly composed of surface- or bulk-diffusion?

Author response:

As described above, we have done a semi-quantitative analysis of the growth process so that better understanding may be achieved. According to the study of the growth kinetics, we propose the growth process is composed of both surface- and bulk-diffusion. As shown in Fig. 1 and Video S1-S3, the growth of alkali metal particles slowed down gradually. At an early stage, oxidation did not occur and the metal growth could be driven by both surface- and bulk-diffusion. We agree with the reviewer's opinion that the growth process could be a competition between irradiation damage and diffusion flux. As the particle grows, irradiation damage becomes more severe and the diffusion flux is depleted. It might be possible that surface diffusion is dominant at this stage. At the late stage in Fig.1 or as the case in Fig.3i and Video S5, once oxidation occurred on those as-formed particles, further illumination could still induce the lithium metal growth from the particles with oxide layers. With the confinement of oxide layers, lithium growth may be dominated by bulk diffusion at this stage.

Author action:

We have added the discussion of diffusion into the main text (Page 8).

3. Although the mechanism for alkali metal production is mainly radiolysis and charging effect, knock-on effect cannot be ignored during the subsequent growth period, especially for light elements such as Li. In my opinion, the growth process is

very likely a competition between irradiation damage and diffusion flux. If the diffusion flux is approximately constant, the knock-on effect gets more and more severe as the metal foil grows, which slows down the growth rate, as observed in Fig. 1. Oxidation may further reduce the growth rate by either confinement (bulk diffusion dominated) or reduction in diffusion flux (surface diffusion dominated).

Author response:

We agree with the reviewer that knock-on damage can not be ignored and the growth process could be a competition between irradiation damage and diffusion flux. Detailed discussion on this point has been provided above in the response to the previous comment.

4. After long-time irradiation, lithium may be depleted while only a lithium oxide shell is left. Thus, the oxide layer thickening mechanism observed in this work could be different from those oxidation process we normally observe.

Author response:

In fact, our observations are in agreement with what people normally observe. The reviewer might misunderstand our description on this point.

In our experiments, the condition of lithium depletion with oxide shell left certainly happened when the beam intensity was relatively high (usually more than $500 \text{ e}\text{\AA}^{-2}\text{s}^{-1}$) and the irradiation time was long enough. This is in agreement with what the reviewer as well as others normally see. However, the oxide layer thickening process in our experiments was carried out in a different situation, that is, the electron beam was off during the time the particles got oxidized. Therefore, the thickening of oxide layer occurred without beam irradiation. We only took one image every 5 min with low dose beam illumination ($\sim 10\text{-}100 \text{ e}\text{\AA}^{-2}\text{s}^{-1}$) and the exposure time of each image is 0.5 s. This is the experiment procedure and imaging parameters used in Supplementary Fig. 10.

Author action:

We have highlighted this point in the figure caption of Fig.S10 to avoid misunderstanding.

5. Can the authors provide any evidence to prove the produced metals are in the form of a foil (as claimed in the manuscript) rather than a nanoparticle? This reflects the growth mode, i.e., 2D- vs 3D-growth.

Author response & action:

In the initial submitted manuscript, we used the word “foil” to describe the as-grown alkali metals. The thickness of these so-called foils has been measured using EELS and shown in Fig.S7 of the initial submission (now as Fig.S9 in the revised manuscript). As can be seen from Supplementary Fig. 9, the thickness of a lithium foil was measured to be around 60 nm while its length is about 100 nm. For the case of a sodium foil measured, the thickness is around 30 nm. Therefore, the so-called foils are thick foils, not the two-dimensional foils. To make it general and to avoid misunderstanding, we have changed the description of “foil” to “particle” in the revised manuscript. And for these particles, we believe the growth mode is 3D-growth.

6. The claimed “layer-by-layer” growth mechanism is not straightforward in Fig. 3.

Author response & action:

We have revised it and made it clear in Fig.3. To better illustrate the growth frontier, different lines are adopted so that the position of the earlier frontier is shown together the current one. In this case, the evolution of the edge and the additive layer could be better seen in each image, as displayed in Fig. 3a-f. One may also see it by referring to Supplementary Video 4. In addition, the growth kinetics of this particle are investigated and the results are shown in Fig. 3g and 3h.

7. In Fig. 4f-h, can the authors specify how is the mapping performed? In Fig. 4d and

4l, are there other elements, such as oxygen, besides alkali metals? Sodium oxide (Fig. 4m) appears differently compared to lithium oxide (Fig. 4e). Have the authors investigated whether sodium oxide is single crystalline or polycrystalline?

Author response:

Fig. 4f-h are false colored Fourier-filtered images. The selected Fourier spots is shown inset in each image. By selecting different Fourier spots in the Fourier transform pattern and making inverse Fourier transform, the corresponding image shows the lattice fringes of selected spots. Using this method, it can be seen that the inset image of Fig.4e contains different sets of lattices of lithium oxide, suggesting polycrystalline feature of the oxide layer.

Fig. 4d and 4l are the EFTEM mapping of lithium and sodium element, respectively. As detailed in the SI, EFTEM image is obtained by selecting certain width of the energy window for a specific element. EFTEM, together with HRTEM, are strong evidence to prove the particles include alkali metal elements. As for the oxygen element, the as-grown particles should not contain too much oxygen before they got sufficiently oxidized, as evidenced by HRTEM in Fig.3. For elemental mapping using EFTEM, we did try to collect oxygen signal as concerned by the reviewer. However, the required time of signal acquisition for oxygen is extremely long as the energy loss of O-K edge is at ~530 eV with very low signal to noise ratio. After the long-time beam irradiation (more than 30 seconds), the particles had already been damaged. As a consequence, EFTEM map of oxygen element could not be obtained. This missing signal of oxygen is also in agreement of our HRTEM results (Fig.3) that the as-grown particles are pure alkali metals. To be noted that, after they got sufficiently oxidized, the oxygen signal could be detected. The oxygen signal could be recognized in STEM-EDS map (Supplementary Fig.12) and STEM-EELS map (Supplementary Fig.13) in the fully oxidized samples. And of course the samples were damaged after STEM mapping.

The difference of lithium oxide (Fig. 4e) and sodium oxide (Fig. 4m) comes from different time the alkali particles placed in the TEM column. The longer the particles

placed in the column (without beam illumination), the thicker the oxide layers. In Fig.4, the sodium particles were placed in the column shorter than lithium particles, therefore the oxide layers were thinner than that of lithium. Generally speaking, both lithium and sodium oxide layers are polycrystalline. Specific to the cases shown in Fig.4, sodium oxide shows uniform orientation in a small area as in the HRTEM image of Fig.4m. But on a larger scale, sodium oxide is also polycrystalline as the case of lithium oxide. This can be also verified from the polycrystalline rings in Fig. 2l.

Author action:

Related discussions have been added into the SI.

8. The vacuum inside the TEM (10^{-5} Pa) is not quite qualified for ultrahigh vacuum as mentioned in the abstract.

Author response:

We have corrected it, changing “ultra-high vacuum” to “high vacuum” which usually refer to the vacuum condition in the range 100mPa to 100nPa.

Response to Reviewer 3

Reviewer #3 (Remarks to the Author):

In their manuscript entitled “Imaging the In situ Production of Alkali Metals at High Spatiotemporal Resolution” C. Liang, X. Zhang, S. Xia, Z. Wang, J. Wu, B. Yuan, X. Luo, W. Liu, W. Liu and Y. Yu report on detailed investigation of leaching and crystallization of alkali metals - lithium and sodium - from their corresponding salts (fluorides and carbonates) upon exposure to electron beam. Crystallization of alkali metals followed by formation of thin oxide layer was observed even under ultra-high vacuum conditions and was extensively studied by various electron microscopy techniques (diffraction, atomic resolution imaging, EELS and EDX). Various factors, such as low and moderate temperatures, type of TEM grid were considered to

elucidate the mechanism of reduction of alkali metals by electron beam.

This is a well-rounded study of the chemical phenomena occurring under very special conditions (electron beam and ultra-high vacuum). While this is an interesting study, which may contribute to our understanding of electron beam interactions with matter, I do not see how this study “may lead to a more complete understanding of the structure and property 216 of Group I elements,” as claimed by Authors. Overall, I can recommend this paper to be published in Nature Communications, but urge Authors to eliminate unsupported claims from the manuscript.

Author response:

We highly appreciate the reviewer’s comments on our manuscript. In the revised manuscript, we have deleted the unsupported claims as suggested. Moreover, we have done substantial amount of additional experiments to strengthen the conclusions. As an application, we demonstrated that our method enabled the in-situ study of the physical contact property of lithium metals (Fig. 6). Major changes are listed below:

- (1) In the revised manuscript, the title has been changed to “Unravelling the room-temperature atomic structure and growth kinetics of lithium metal”.
- (2) Fig. 3 has been updated with a detailed analysis of the growth of lithium particles. Besides, we have demonstrated the growth of lithium dendritic whiskers. Quantitative investigation of the whisker growth kinetics here has been compared with the electrical current-driven growth of lithium dendrites in lithium ion batteries.
- (3) Fig. 5 has been removed. We have updated a new Fig.5, demonstrating an immediate advantage of our study is that, ambiguities in lithium metal growth on garnet-type solid electrolytes of lithium ion batteries could be solved. We employed our method on $\text{Li}_7\text{La}_3\text{Zr}_2\text{O}_{12}$ (LLZO) and our observations indicated that previous reported lithium metal growth on LLZO could be an artifact and the true origin came from surface contaminated lithium carbonates.
- (4) Fig.6, i.e., a new figure has been added, demonstrating an application of our method is that, the physical contact property of lithium metal and surface

passivation oxide layer could be studied. By introducing an AFM tip inside the TEM, we were able to guide the growth of lithium whisker and study the contact property of Li-Li, Li₂O-Li₂O, and Li-Li₂O, respectively. These results may provide better understanding of lithium dendrite growth and solid electrolyte interphase (SEI) related issues in batteries.

Updated figures have been provided in the revised manuscript as well as in the response to Reviewer #1 and Reviewer #2 as shown above.

REVIEWER COMMENTS

Reviewer #2 (Remarks to the Author):

The authors responded the comments carefully and revised the manuscript to a great extent. Some new comments should be considered.

It was said that the observations unravel the ambiguities in lithium metal growth on garnet-type solid electrolytes for lithium-metal batteries. What is the relationship between the Li growth caused by electron beam irradiation and the growth on garnet-type solid electrolytes? What can we learn from the observation?

In fig.3, the growth length versus time was present in fig 3g and 3f. A dramatic increasing can be found in [1-10] direction. Meanwhile, two types of Li whisker growth can be found in fig3i-j. The authors should give a convictive explanation why these phenomena could occur, that is what is the physical picture?

Reviewer #3 (Remarks to the Author):

Authors have considerably revised the manuscript, added additional figures, and deleted unsupported claims. I recommend this manuscript publishing this manuscript in Nature Communications.

Response to Reviewers' Comments for Manuscript NCOMMS-19-33196B

Response to Reviewer 2

Reviewer #2:

Comments:

The authors responded the comments carefully and revised the manuscript to a great extent. Some new comments should be considered.

Author response:

We highly appreciate the reviewer's positive comments and our responses to the specific comments are provided below.

1. It was said that the observations unravel the ambiguities in lithium metal growth on garnet-type solid electrolytes for lithium-metal batteries. What is the relationship between the Li growth caused by electron beam irradiation and the growth on garnet-type solid electrolytes? What can we learn from the observation?

Author response:

Lithium metal growth from garnet-type solid electrolytes such as $\text{Li}_7\text{La}_3\text{Zr}_2\text{O}_{12}$ (LLZO) under the electron beam irradiation has been reported by different research groups (Refs. 26 and 27). The observations were carried out using SEM in both cases. Here, using TEM, we study the same topic as in Refs. 26 and 27. The advantage is that our observations using TEM enable a better spatial resolution compared to SEM (Refs. 26 and 27), and we found that the reported "Li growth on garnet-type solid electrolyte" could be an illusion. High resolution observations indicated that Li was grown from the surface contamination lithium carbonate, not from the garnet-type solid electrolyte.

The key point we can learn from our observations is that the true origin of Li growth is from surface contamination of lithium carbonates, not the garnet-type solid electrolyte. As limited by spatial resolution and the lack of lithium elemental detection capability (Li cannot be detected in EDS) in SEM, it is difficult to find out the true origin of Li growth, as in Refs. 26 and 27. Therefore, the previous SEM observations came to the conclusion that Li can be grown from LLZO under the electron beam irradiation. In contrast, as demonstrated in this work, with high spatial resolution imaging and electron diffraction in TEM and the Li elemental analysis capability using EELS, we were able to study the Li growth carefully and our observations indicated that the true origin of Li growth is from the surface contamination of lithium carbonate (Figs. 5a and 5b), which is easily formed on the garnet surface when exposed to water and carbon dioxide in the air (Ref. 28). Electron beam irradiation on pure LLZO could not form Li (Figs. 5c and 5d).

Apart from the points mentioned above, it can also be seen that careful and comprehensive characterizations are needed for electron beam sensitive materials to avoid possible illusions.

Author action:

We have added discussions into the revised manuscript (Page 12-13).

2. In fig.3, the growth length versus time was present in fig 3g and 3f. A dramatic increasing can be found in [1-10] direction. Meanwhile, two types of Li whisker growth can be found in fig3i-j. The authors should give a convictive explanation why these phenomena could occur, that is what is the physical picture?

Author response:

We take the nice suggestions from the reviewer and provide explanations to these phenomena.

- (1) For the dramatic increasing growth in [1-10] direction, it could be understood from the perspective of surface energy and mass transport. With the continuous

growth along [10-1] direction (Figs. 3b and 3c), the area of (1-10) facet and its surface energy increases. For 3D growth of the lithium particle, as known from the Wulff theorem [1], minimization of the overall surface energy determines the final morphology and it is possible that continuous growth of the (10-1) facet be suppressed while other equivalent {110} faces grow one after the other. Under this circumstance, the particle tends to change the growth direction. In terms of the different growth rate between [10-1] and [1-10] direction, it is related with the anisotropic mass transfer along these two directions [1]. The rate of mass transfer could be determined by the decomposition of the matrix of the lithium salts. Anisotropic decomposition of the lithium salts may provide anisotropic diffusion flux of lithium source along different directions. The fast growth along [1-10] direction could be attributed to the sudden and large amount of diffusion flux provided. The accumulated lithium source was in the status of supersaturation, and therefore the growth along [1-10] direction could quickly initiate.

- (2) We have observed the growth of lithium whiskers apart from lithium particles, indicating 1D growth occurred under certain circumstance. The 1D- or 3D-growth mode is on one hand related to the confinement from surface passivation layers, and on the other hand related to the diffusion flux which is dependent of the strength of the driving force (electron beam irradiation). 1D-growth could occur at the pinhole of the surface passivation layers of as-formed lithium particles (Type I whisker) or at some unique nucleation sites of the lithium salts where the diffusion is confined (Type II whisker). The different growth kinetics of two types of whisker could be related with different diffusion barriers in lithium metal and lithium salts [2]. The accelerated growth of Type I whisker is in agreement with the case of lithium whisker growth in the liquid electrolyte of lithium ion batteries (growth stage II in Ref. 22), and uniform growth of Type II lithium whiskers suggests the decomposition reaction of lithium salts occurs uniformly, and the formation of lithium metal may be reaction-rate limited in this case (one may also refer to Fig.5 in Ref.

21).

Reference:

[1] Springer Handbook of Crystal Growth (Springer, 2010)

[2] Ozhabes, Y., Gunceler, D. & Arias, T. A. Stability and surface diffusion at lithium-electrolyte interphases with connections to dendrite suppression. arXiv: 1504.05799v1 (2015)

Author action:

We have added discussions and references into the revised manuscript (Page 7-8).

Response to Reviewer 3

Reviewer #3:

Comments:

Authors have considerably revised the manuscript, added additional figures, and deleted unsupported claims. I recommend this manuscript publishing this manuscript in Nature Communications.

Author response:

We highly appreciate the reviewer's positive comments.

REVIEWERS' COMMENTS

Reviewer #2 (Remarks to the Author):

My concerns have been addressed in the revisions and it's ready to publish in Nature Communications.

Response to Reviewers' Comments for Manuscript NCOMMS-19-33196B

Response to Reviewer 2

Reviewer #2:

Comments:

My concerns have been addressed in the revisions and it's ready to publish in Nature Communications.

Author response:

We highly appreciate the reviewer's positive comments.